# Exploring Korean Middle- and Old-Aged Citizens’ Subjective Health and Quality of Life

**DOI:** 10.3390/bs12070219

**Published:** 2022-06-29

**Authors:** Joonho Moon, Won Seok Lee, Jimin Shim

**Affiliations:** 1Department of Tourism Administration, Kangwon National University, Chuncheon 24341, Korea; joonhomoon0412@gmail.com; 2Department of Tourism and Recreation, Kyonggi University, Seoul 03746, Korea; shim5515@naver.com

**Keywords:** middle to old-aged citizen, quality of life, subjective health, travel, cultural activity, economic activity

## Abstract

The goal of this research is to investigate the determinants of subjective health and quality of life with a particular focus on middle- and old-aged citizens. Subjective health is an antecedent of quality of life. For both attributes, travel frequency, economic activity, and cultural activity frequency are the main explanatory variables. Korean middle- and old-aged citizen research panel data was used to derive the data; the study periods are 2008, 2010, 2012, 2014, and 2016. The present work used an econometric method to analyze this panel data. The results show that subjective health positively affects quality of life; meanwhile, economic activity positively affects both subjective health and quality of life. It is also found that cultural activity and travel exert inverted U-shape impacts on subjective health and quality of life. The control variables in this research were gender, body mass index, birth year, and personal assets. These results could help guide policy makers in designing more efficient welfare policies for middle- and old-aged citizens.

## 1. Introduction

Statistics Korean [1] forecasted that Korean society will soon be confronted with a population decrease (as the population, which was 51.8 million in 2021, is projected to be 47.7 million in 2050) with a substantially increased proportion of elderly people (16.5% in 2021 and 40.4% in 2050). Such an aging trend leads to problems such as decreasing productivity, increasing medical costs, and decreasing tax income [2,3,4]. In this context, it is important to elucidate the characteristics of the lives of middle and old aged senior citizens to build more adequate government policy. 

The main explained attributes of this work are quality of life and subjective health. Both attributes have been commonly explored by scholars because the elements represent the mental and physical status of an individual as well as their overall happiness in living [5,6,7,8,9]. As the candidate variables, this research selected travel frequency, economic activity, and cultural activity. Daily life in old age tends to involve more spare time; it is therefore vital to determine how this time can be used for a better life. Senior citizens are more likely to enjoy travel and cultural activities because they have more fertile opportunities for leisure than younger generations. Prior studies have addressed that career break impairs both health condition and quality of life in old age [10,11]. This indicates that continuing economic activity is likely to improve one’s overall life condition because it can reduce their feeling of loss. 

The main theoretical background of this work is the law of diminishing marginal utility. According to the law of diminishing marginal utility, highly frequent goods consumption reduces utility, which in turn results in negative utility because of the huge cost [12,13,14]. This could be applied to the frequency of travel and cultural product in the sense that the high frequency of goods consumption causes negative utility [15,16]. Travel and cultural activity could be considered as a service product that is intangible, inseparable, and perishable [17,18,19]. The law of diminishing marginal utility can be applied to service products. Thus, this study specifically adds to the literature by ensuring the accountability of the law of diminishing marginal utility for a service product. 

All things considered, the purpose of this research is to investigate the influential attributes on subjective health and quality life of Korean middle and old aged senior citizens. This study also intends to provide information to help policy makers establish more adequate systems for middle and old aged senior citizens’ welfare. 

## 2. Review of Literature and Hypotheses Development

### 2.1. Quality of Life

Quality of life refers to individuals’ self-evaluations about their life conditions [9,20]. Quality of life measurements aim to integrate all aspects of life [6,7]. Numerous studies have used quality of life as the main variable of interest. For instance, Kim et al. [21] investigated the effect of tourism on quality of life. In another study with participants from Thailand, Senasu and Singhapakdi [22] used quality of life as the central element. Meanwhile, Drakouli et al. [23] used quality of life as the dependent variable when studying children. Hung et al. [24] similarly reported the determinants of quality of life by scrutinizing cancer patients. In work related to the present topic, Alexiou et al. [25] explored influential variables on quality of life among elderly people. 

### 2.2. Subjective Health

Subjective health refers to how individuals perceive their own health conditions, and it includes both physical and mental aspects together [8,26]. Prior research has tested subjective health as an explained attribute. For example, Petanidou et al. [27] demonstrated the determinants of subjective health in a study considering Greek adolescents. In a similar vein, Boarini et al. [5] revealed significant variables that account for the subjective health of participants from Organization for Economic Cooperation and Development (OECD) countries. In another study, Bloem et al. [28] studied the antecedents of subjective health using health care consumers as the study subject. Scholars have also demonstrated the effect of subjective health on quality of life by showing that healthy mental and body condition is the basis for better life quality [29,30]. Meanwhile, Chou et al. [31] showed that quality of life is positively affected by the subjective health of patients. In a meta-analysis, Degnan et al. [32] found that subjective health exerts a positive effect on quality of life. In a study with homeless participants, Gadermann et al. [33] showed that subjective health had a positive effect on quality of life. Given this literature review, the present study proposes the following research hypothesis.

**Hypothesis** **1:**
*Subjective health has a significant positive effect on quality of life.*


### 2.3. Travel

Travel helps people have better health conditions and improved life quality because it allows people to stray away from their daily lives and have exotic and fruitful experiences [34,35,36,37]. Indeed, fertile studies have offered empirical evidence showing that travel leads to better health and life outcomes, which could become the most crucial motivation for travel for certain people. Dolnicar et al. [38] stated that travel experience improves quality of life. Hernández-Mogollón et al. [39] demonstrated that travel had a positive effect on quality of life among a sample in Spain. Backer [40] also found that quality of life is positively affected by traveling. In the case of subjective health, De Vos et al. [41] contended that travel plays a pivotal role in promoting subjective health conditions. Yu et al. [42] also demonstrated that holiday leisure travel significantly enhances subjective health condition.

### 2.4. Economic Activity

People create, achieve, and develop their life goals and attain earnings through economic activities [43,44]. In such economic activities, individuals understand their own presence; this understanding leads to healthier and better living [45,46,47]. Yu and Choe [48] documented that economic activity has a positive effect on subjective health among workers with disabilities. Stevenson and Wolfers [49] argued that individuals become healthier by economic participation through interactions with others and financial rewards. Diener et al. [50] showed that economic participation is positively related to subjective health. In another study, Tvaronavičienė et al. [51] presented that the life quality of youth is determined by employment. Namazi et al. [52] also reviewed the literature and concluded that quality of life is positively affected by economic participation. In addition, Eum and Kim [53] inspected Korean elderly people and disclosed a positive impact of economic participation on living quality. 

### 2.5. Cultural Activity

Cultural activities include music, art performances, exhibitions, concert, musical, etc. [54,55,56]. Scholars have shown that cultural activities are essential in promoting health conditions and improving quality of life by providing participants with positive mental energy. Brajša-Žganec et al. [57] and Wheatley and Bickerton [58] asserted that cultural activities are useful for enhancing subjective health. Västfjäll et al. [59] stated that cultural activities exert positive effects on subjective health. In terms of quality of life, Hong et al. [60] addressed that cultural activity, such as visiting museums and exhibitions, improves quality of life because individuals broaden their horizons by experiencing various masterpieces. Coffman [61] showed that the quality of life of elderly people is positively impacted by music. Cooke et al. [62] and Abdulah and Abdulla [63] also found that art plays an imperative role in enhancing the life quality of patients.

### 2.6. Law of Diminishing Marginal Utility

The law of diminishing marginal utility also argues that marginal utility through additional consumption declines with more consumption [15,16]. The law of diminishing marginal utility has been widely used as a theoretical underpinning in various studies. Easterlin [64] and Goetz [65] examined the law of diminishing marginal utility using income and water resources, respectively. Tan and Zhang [12] demonstrated the explanatory power of the law of marginal utility in the area of the wireless network service sector. Line et al. [13] and Liu et al. [14] confirmed the accountability of the law of diminishing marginal utility in intangible product sectors such as the restaurant and hotel industries. In general, the research shows that cultural activity and travel play pivotal roles in improving subjective health and quality of life. However, excessive travel and cultural activity are likely to show reduced marginal utility. Further, the cost associated with travel and cultural activity could exceed the utility gained by the lavish consumption of travel and cultural products. This implies that a curvilinear (inverted-U) shape effect could be anticipated in terms of the frequency attributes. With this background, this study proposes the following research hypotheses:

**Hypothesis** **2a:**
*Travel frequency has a significant curvilinear (inverted U-shaped) effect on quality of life.*


**Hypothesis** **2b:**
*Travel frequency has a significant curvilinear (inverted U-shaped) effect on subjective health.*


**Hypothesis** **3a:**
*Economic activity has a significant positive effect on quality of life.*


**Hypothesis** **3b:**
*Economic activity has a significant positive effect on subjective health.*


**Hypothesis** **4a:**
*Cultural activity frequency has a significant curvilinear (inverted U-shaped) effect on quality of life.*


**Hypothesis** **4b:**
*Cultural activity has a significant curvilinear (inverted U-shaped) effect on subjective health.*


## 3. Method

### 3.1. Data Collection and Measurement of Variables

This research uses archival data. The data was attained from Korean Senior citizen research panel data (Korean longitudinal study of aging) that is collected by the Korea Employment Information Service. The data defines senior citizens as Koreans who are older than 45 years old. Therefore, the scope of survey participants are middle and old aged senior citizens. The Korea Employment Information Service has performed this survey every two years since 2006. The survey data has been used by several prior studies [66,67,68,69]; such works could ensure the quality of the data. The study period of this research is between 2008–2016. Namely, longitudinal data taken over five years (2008, 2010, 2012, 2014, 2016) was used for the data analysis. The number of participants was 7486. Thus, the number of original observations was 37,430 (7486 × 5). Then, after missing data (165 observations) were eliminated from the sample, 37,375 observations were ultimately valid for data analysis.

The explained variables in this research are quality of life (QOL) and subjective health (SHE). These are measured by point values ranging from 0 to 100. The other variables are the frequency of annual traveling (TRF) and cultural life (CAF): movie, musical, exhibition, sports, and etc. Economic activity is measured by a binary variable (0 = No, 1 = Yes). This work also selects four control variables: body mass index (kg/m^2^) (BMI) birth year (BYR), gender (Male is given Gender = 0; Otherwise, women are given Gender = 1) (GEN), and individual total assets (Unit is ten thousand KRW) (AST). As of March 2022, the currency rate for 1,200 KRW is approximately equivalent to 1 USD. Table 1 depicts the description of variable. 

### 3.2. Data Analysis and Research Model

STATA 13 was used for the statistical package for data analysis. This study computed mean, standard deviation (SD), minimum, and maximum as the descriptive statistics. Then, correlation matrix analysis was conducted to examine the relationships between variables. To test the hypotheses, this study performed the following panel regression methods: ordinary least square (OLS), fixed effect (FE), and random effect (RE). This study ensured the consistency of the coefficients across three results of multiple regression analysis. The fixed effects model incorporates a multiple year dummy into the regression model to minimize omitted variable bias in the panel data, while random effect adds an unobserved effect into the model for estimation [70,71]. This study also implemented quadratic regression analysis, which incorporates a square term into the regression model to attain a point to maximize the explained attributes [71,72]. Plus, this research carried out two stage least square regression (2SLS) model not only to minimize the bias in simultaneous regression equation regarding SHE variables but also to ensure the robustness [70]. This study executed Chow-test to select better model between OLS and FE (H_0_: No difference between OLS and FE) and Hausman test to choose better model between FE and RE (H_0_: No difference between RE and FE) [70,71,72]. To attain the point for maximization, this study carried out differentiation for the first order condition. According to Wooldridge [72], the estimation of quadratic multiple regression model is likely to be undermined by multi-collinearity. To reduce the likelihood of bias by multi-collinearity, this study calculated the variation inflation factor (VIF), which is computed by 1/(1-R^2^). Altogether, Figure 1 describes the research model.

We present the following regression equation:SHE_it_ = *β*_0_ + *β*_1_*TRF*_*it*_ + *β*_2_*TRF*_*it*_^2^ + *β*_3_*EAC*_*it*_ + *β*_4_*CAF*_*it*_ + *β*_5_*CAF*_*it*_^2^ + *β*_6_*GEN*_*it*_ + *β*_7_*BMI*_*it*_ + *β*_8_*BYR*_*it*_ + *β*_9_*AST*_*it*_ + *ε*_*it*_


*QOL*_*it*_ = *β*_0_ + *β*_1_*SHE*_*it*_ + *β*_2_*TRF*_*it*_ + *β*_3_*TRF*_*it*_^2^ + *β*_4_*EAC*_*it*_ + *β*_5_*CAF*_*it*_ + *β*_6_*CAF*_*it*_^2^ + *β*_7_*GEN*_*it*_ + *β*_8_*BMI*_*it*_ + *β*_9_*BYR*_*it*_ + *β*_10_*AST*_*it*_ + *ε*_*it*_


Note: i: *i*th participants, t: *t*th year.

## 4. Results

### 4.1. Descriptive Statistics and Correlation Matrix

The respective means of QOL and SHE are 62.70 and 58.74 and the standard deviations are 15.91 and 19.68, respectively. For EAC and GEN, the means are 0.40 and 0.57, respectively. Table 2 also presents the information on TRF (mean = 1.34, SD = 2.73), CAF (mean = 0.75, SD = 2.19), BMI (mean = 23.26, SD = 2.73), BYR (mean = 1947.93, SD = 10.31), and AST (mean = 21251.41, SD = 31881.82).

Table 3 presents the results of the correlation matrix. QOL is positively correlated with SHE (r = 0.625, *p* < 0.05), TRF (r = 0.154, *p* < 0.05), EAC (r = 0.194, *p* < 0.05), CAF (r = 0.177, *p* < 0.05), BMI (r = 0.042, *p* < 0.05), BYR (r = 0.234, *p* < 0.05), and AST (r = 0.170, *p* < 0.05). SHE is also positively correlated with TRF (r = 0.163, *p* < 0.05), EAC (r = 0.297, *p* < 0.05), CAF (r = 0.194, *p* < 0.05), BMI (r = 0.069, *p* < 0.05), BYR (r = 0.366, *p* < 0.05), and AST (r = 0.152, *p* < 0.05). TRF is positively correlated with EAC (r = 0.101, *p* < 0.05), CAF (r = 0.263, *p* < 0.05), BYR (r = 0.186, *p* < 0.05), and AST (r = 0.099, *p* < 0.05). EAC is also positively correlated with CAF (r = 0.108, *p* < 0.05), BYR (r = 0.035, *p* < 0.05), and AST (r = 0.053, *p* < 0.05).

### 4.2. Results of Hypotheses Testing

Table 4 presents the results of the hypotheses testing with regard to subjective health. Model 1, Model 2, Model 3, and Model 4 are OLS, FE, RE, and 2SLS, respectively. EAC positively affects SHE (β = 4.54, *p* < 0.05). Next, TRF (β = 0.79, *p* < 0.05), TRF^2^ (β = −0.02, *p* < 0.05), CAF (β = 1.10, *p* < 0.05), and CAF^2^ (β = −0.03, *p* < 0.05) significantly account for SHE. Using the coefficients, this research performed differentiation and the first-order condition was computed to obtain the point to maximize SHE (TRF = 19.7, CAF = 18.3). GEN is found to be negatively associated with SHE (β = −1.93, *p* < 0.05); meanwhile, BMI (β = 0.22, *p* < 0.05), BYR (β = 0.47, *p* < 0.05), and AST (β = 0.01, *p* < 0.05) are positively related to SHE. The results are consistent across all three models (Models 1 through 4). The results of the Chow test indicate no difference between OLS and FE (F = 0.00, *p* > 0.05). Also, the Hausman test results imply that there is no difference between RE and FE (χ^2^ = 0.00, *p* > 0.05). All things considered, OLS is regarded as the most adequate model (Model 1).
Model: SHE_it_ = *β*_0_ + *β*_1_*TRF*_*it*_ + *β*_2_*TRF*_*it*_^2^ + *β*_3_*EAC*_*it*_ + *β*_4_*CAF*_*it*_ + *β*_5_*CAF*_*it*_^2^ + *β*_6_*GEN*_*it*_ + *β*_7_*BMI*_*it*_ + *β*_8_*BYR*_*it*_ + *β*_9_*AST*_*it*_ + *ε*_*it*_

Table 5 lists the results of the hypotheses testing for quality of life. As presented in the table, SHE (β = 0.47, *p* < 0.05) and EAC (β = 0.49, *p* < 0.05) exerted positive effects on QOL, TRF (β = 0.49, *p* < 0.05), TRF^2^ (β = −0.01, *p* < 0.05), CAF (β = 0.37, *p* < 0.05), and CAF^2^ (β = −0.01, *p* < 0.05). Given the coefficients, this study carried out differentiation and the first-order condition was calculated to obtain the point to maximize QOL (TRF = 24.5, CAF = 18.5). The results also show that BYR (β = −0.02, *p* < 0.05) is negatively related to QOL, whereas GEN (β = 0.41, *p* < 0.05) and AST (β = 0.01, *p* < 0.05) are positively related to QOL. All VIF values are less than 10, indicating that the model is less likely to be biased by multicollinearity, as listed in Table 4 (range: 1.02–3.51) and Table 5 (range: 1.02–3.52). The results were consistent in all three econometric models (Model 5 through Model 8). To summarize, all hypotheses are supported by the results of the multiple linear regression analysis. The Chow test results show no difference between OLS and FE (F = 0.00, *p* > 0.05). In addition, the results of the Hausman test present no difference between RE and FE (χ^2^ = 0.00, *p* > 0.05). This indicates that OLS is the most appropriate (Model 5).
Model: *QOL*_*it*_ = *β*_0_ + *β*_1_*SHE*_*it*_ + *β*_2_*TRF*_*it*_ + *β*_3_*TRF*_*it*_^2^ + *β*_4_*EAC*_*it*_ + *β*_5_*CAF*_*it*_ + *β*_6_*CAF*_*it*_^2^ + *β*_7_*GEN*_*it*_ + *β*_8_*BMI*_*it*_ + *β*_9_*BYR*_*it*_ + *β*_10_*AST*_*it*_ + *ε*_*it*_

## 5. Discussion

This work aimed to examine the influential attributes of quality life and subjective health for Korean middle- and old-aged citizens. The results suggest that improving subjective health is essential for improving quality of life. The results also demonstrated that the frequency of travel and cultural activities both exerted inverted U-shaped effects. This can be inferred to mean that too high a frequency of travel and cultural activities could diminish the associated utility gained by middle- and old-aged citizens, while simply costing them time and effort to participate in travel and cultural activities. In other words, the repeated consumption of travel and cultural products reduces utility over time. Therefore, maintaining an optimal frequency of travel and cultural activities is imperative for subjective health and life quality. The results also showed that economic activity enhances both subjective health and quality of life for middle- and old-aged citizens. It implies that jobs for middle- and old-aged citizens in Korea are more worthwhile for accomplishing better life and health conditions. In terms of subjective health, women showed a lower level of subjective health, and body mass index improved subjective health. Meanwhile, quality of life among females was better than that among males. Namely, women had a better quality of life and worse subjective health than men. Further, younger senior citizens have better subjective health and quality of life. Finally, possessing more personal assets is critical for enhancing both subjective health and quality of life for Korean middle- and old-aged citizens.

## 6. Conclusions

### 6.1. Theoretical and Practical Implications

This study makes important theoretical contributions: Above all, this research ensured the accountability of the law of marginal utility decline in the areas of quality of life and subjective health by using travel and cultural life as the explanatory attributes. Specifically, this work demonstrates that travel and cultural activity frequency both have curvilinear effects on subjective health and quality of life by researching Korean middle and old aged senior citizens. Moreover, the results of this research externally validate the findings of prior research. In particular, this research confirmed the link between subjective health and quality of life in the context of Korean middle and old aged senior citizens [31,33]. The current study also confirmed the impacts of economic participation on subjective health and quality of life in the case of Korean middle and old aged senior citizens [50,51]. In sum, this work externally validated the outcomes of prior studies by offering significant association between attributes.

This work has practical implications. First, this information could be used by policy makers. Specifically, a government budget could be allocated for creating jobs for middle- and old-aged citizens. Financial resources could also be allocated to education for improving middle- and old-aged citizens’ job skills. Second, it could be necessary for policy makers to invest in travel and cultural destinations. However, careless spending should be avoided because middle- and old-aged citizens’ marginal utility could be decreased with repetitive activities. Hence, the costs of travel and cultural programs would need to be considered by policy makers because the resources of middle- and aged citizens are constrained. In terms of the results of the control variables, government policy for subjective health and quality of life needs to be differentiated depending on gender. For example, policies related to better life quality should focus more on males, whereas policies related to improving health should focus more on females. In addition, policy makers should contemplate how to enhance both the subjective health and the quality of life of older senior citizens. Government resources should also be dedicated to improving the health of middle- and old-aged citizens. This could be achieved through various channels, for example, offering medical support and healthy food while also addressing the isolation of some citizens.

### 6.2. Limitations

This study has limitations. First, the data were only available up until 2016. Future research might be able to use more updated information because pre-pandemic data might not be sufficient to offer a real picture of the current situation. In particular, the period of data in future studies should include 2020, which would allow researchers to investigate the effects of COVID-19 on senior citizens. This research also depended on archival data. Future studies should consider more advanced items for the measurement of the attributes in this research. Furthermore, it is possible that there could be a more influential attribute to account for both subjective health and quality of life. Future research thus needs to search for more influential variables because the R-square of this research could be improved. Such an effort might lead to more useful future research.

## Figures and Tables

**Figure 1 behavsci-12-00219-f001:**
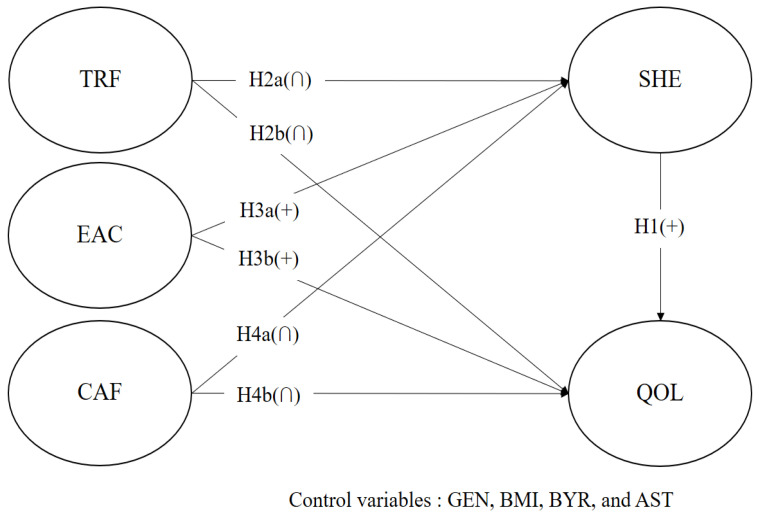
Research model. Note: quality of life (QOL), subjective health (SHE), travel frequency (TRF), economic activity (EAC), cultural activity frequency (CAF), gender (GEN), body mass index (BMI), birth year (BYR), personal assets (AST).

**Table 1 behavsci-12-00219-t001:** Variable description.

Name	Code	Description (Unit)
Quality of life	QOL	(0 = Very poor, 100 = Very good)
Subjective health	SHE	(0 = Very poor, 100 = Very good)
Travel frequency	TRF	Annual travel frequency (times)
Economic activity	EAC	(0 = No, 1 = Yes)
Cultural activity frequency	CAF	Annual art cultural activity participation frequency (times)
Gender	GEN	(0 = Male, 1 = Female)
Body mass index	BMI	Body mass index of survey participants
Birth year	BYR	Birth year of survey participants
Personal assets	AST	Personal assets (10 thousand KRW)

Note: KRW denotes Korean won.

**Table 2 behavsci-12-00219-t002:** Descriptive statistics (N = 37375).

Variable	Mean	SD	Minimum	Maximum
QOL	62.70	15.91	0	100
SHE	58.74	19.68	0	100
TRF	1.34	2.73	0	50
EAC	0.40	0.49	0	1
CAF	0.75	2.19	0	40
GEN	0.57	0.49	0	1
BMI	23.26	2.73	12.11	43.03
BYR	1947.93	10.31	1915	1963
AST	21,251.41	31,881.82	10	507,209.90

Note: SD denotes standard deviation. quality of life (QOL), subjective health (SHE), travel frequency (TRF), economic activity (EAC), cultural activity frequency (CAF), gender (GEN), body mass index (BMI), birth year (BYR), personal assets (AST).

**Table 3 behavsci-12-00219-t003:** Correlation matrix.

Variable	1	2	3	4	5	6	7	8
1.QOL	1							
2.SHE	0.625 *	1						
3.TRF	0.154 *	0.163 *	1					
4.EAC	0.194 *	0.297 *	0.101 *	1				
5.CAF	0.177 *	0.194 *	0.263 *	0.108 *	1			
6.GEN	−0.058 *	−0.111 *	−0.024 *	−0.259 *	0.029 *	1		
7.BMI	0.042 *	0.069 *	0.033 *	0.035 *	0.010 *	0.020 *	1	
8.BYR	0.234 *	0.366 *	0.186 *	0.486 *	0.264 *	−0.036 *	0.111 *	1
9.AST	0.170 *	0.152 *	0.099 *	0.053 *	0.157 *	−0.163 *	0.022 *	0.067 *

Note: * *p* < 0.05, quality of life (QOL), subjective health (SHE), travel frequency (TRF), economic activity (EAC), cultural activity frequency (CAF), gender (GEN), body mass index (BMI), birth year (BYR), personal assets (AST).

**Table 4 behavsci-12-00219-t004:** Results of hypotheses testing.

Variable	Model 1 (OLS) β (*t*-Stat)	Model 2 (FE) β (*t*-Stat)	Model 3 (RE) β (Wald)	Model 4 (2SLS) β (*t*-Stat)	VIF
Intercept	−866.93 (−36.87) *	−866.93 (−36.87) *	−866.93 (−36.87) *	−866.93 (−36.87) *	
TRF	0.79 (12.10) *	0.79 (12.10) *	0.79 (12.10) *	0.79 (12.10) *	3.51
TRF2	−0.02 (−7.54) *	−0.02 (−7.54) *	−0.02 (−7.54) *	−0.02 (−7.54) *	3.22
EAC	4.54 (19.09) *	4.54 (19.09) *	4.54 (19.09) *	4.54 (19.09) *	1.43
CAF	1.10 (13.90) *	1.10 (13.90) *	1.10 (13.90) *	1.10 (13.90) *	3.49
CAF2	−0.03 (−8.22) *	−0.03 (−8.22) *	−0.03 (−8.22) *	−0.03 (−8.22) *	3.04
GEN	−1.93 (−9.22) *	−1.93 (−9.22) *	−1.93 (−9.22) *	−1.93 (−9.22) *	1.12
BMI	0.22 (6.18) *	0.22 (6.18) *	0.22 (6.18) *	0.22 (6.18) *	1.02
BYR	0.47 (38.79) *	0.47 (38.79) *	0.47 (38.79) *	0.47 (38.79) *	1.49
AST	0.01 (17.97) *	0.01 (17.96) *	0.01 (17.97) *	0.01 (17.97) *	1.07

*F*-value	717.28 *	795.01 *	-	717.28 *	
Wald χ^2^	-	-	6455.56 *	-	
R2	0.1746	0.1746	0.1746	0.1746	

Note: Dependent variable: SHE, * *p* < 0.05, FE denotes fixed effects, RE denotes random effects, 2SLS stands for two-stage least squares, VIF stands for variation inflation factor, optimal frequency to maximize QOL: Δ/ΔTRF = 19.7, Δ/ΔCAF = 18.3. Quality of life (QOL), subjective health (SHE), travel frequency (TRF), economic activity (EAC), cultural activity frequency (CAF), gender (GEN), body mass index (BMI), birth year (BYR), personal assets (AST).

**Table 5 behavsci-12-00219-t005:** Results of hypotheses testing.

Variable	Model 5 (OLS) β (*t*-Stat)	Model 6 (FE) β (*t*-Stat)	Model 7 (RE) β (Wald)	Model 8 (2SLS) β (*t*-Stat)	VIF
Intercept	82.64 (4.93) *	82.64 (4.93) *	82.64 (4.93) *	82.64 (4.93) *	
SHE	0.47 (119.21) *	0.47 (119.21) *	0.47 (119.21) *	0.47 (119.21) *	1.21
TRF	0.49 (10.63) *	0.48 (10.63) *	0.48 (10.63) *	0.49 (10.63) *	3.52
TRF2	−0.01 (−7.02) *	−0.01 (−7.02) *	−0.01 (−7.02) *	−0.01 (−7.02) *	3.23
EAC	0.56 (3.36) *	0.56 (3.36) *	0.56 (3.36) *	0.56 (3.36) *	1.44
CAF	0.37 (6.75) *	0.37 (6.75) *	0.37 (6.75) *	0.37 (6.75) *	3.51
CAF2	−0.01 (−2.53) *	−0.01 (−2.53) *	−0.01 (−2.53) *	−0.01 (−2.53) *	3.05
GEN	0.41 (2.86) *	0.41 (2.86) *	0.41 (2.86) *	0.41 (2.86) *	1.02
BMI	−0.03 (−1.53)	−0.03 (−1.53)	−0.03 (−1.53)	−0.03 (−1.53)	1.56
BYR	−0.02 (−2.89) *	−0.02 (−2.89) *	−0.02 (−2.89) *	−0.02 (−2.89) *	1.12
AST	0.01 (14.65) *	0.01 (14.65) *	0.01 (14.65) *	0.01 (14.65) *	1.08

*F*-value	1944.04 *	1943.78 *	-	1944.04 *	
Wald χ^2^	-	-	194,408.36 *	-	
R2	0.3891	0.3891	0.3891	0.3891	

Note: Dependent variable: QOL, * *p* < 0.05, FE denotes fixed effects, RE denotes random effects, 2SLS stands for two-stage least squares, VIF stands for variation inflation factor, optimal frequency to maximize SHE: Δ/ΔTRF = 24.5, Δ/ΔCAF = 18.5. Quality of life (QOL), subjective health (SHE), travel frequency (TRF), economic activity (EAC), cultural activity frequency (CAF), gender (GEN), body mass index (BMI), birth year (BYR), personal assets (AST).

## Data Availability

Not applicable.

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
