# Peer review of "Exploring Korean Middle- and Old-Aged Citizens’ Subjective Health and Quality of Life"

_behavsci, 2022, doi:10.3390/bs12070219_

Round 1

Reviewer 1 Report

The work concerns a very interesting problem of the quality of life of seniors in Korea but thework is written in a rather complicated way, difficult to read. 1.     There is no detailed information on the authors' affiliation 2.     In the Review of literature and hypotheses development section, there is, in my opinion, too much information with too many citations, which makes the essence of the topic not very clear 3.     There should be an explanation of the abbreviations below the tables. 4.     The age criteria of a senior is strongly underestimated - a person over 45 was considered a senior. With such a low age considered as a senior, a division into age groups should be made, because the possibilities of, for example, traveling or cultural activity for a 45-year-old and an 80-year-old are completely different. 5.     The discussion is definitely too short, it should include a discussion of the results together with their reference to other researchers 6.     Conclusions, too extensive, some information from conclusions should be included in the discussion  

Reviewer 2 Report

 The study deals with a very topical issue. Despite this, I think it is desirable to clearly define the type of study within the article and refer to the guidelines that can be consulted on The EQUATOR Network | Enhancing the QUAlity and Transparency Of Health Research (equator-network.org)

furthermore, I find that relying on such old, pre-pandemic data does not offer a real picture of the current situation. The conclusions can therefore be shared but not in the current state of things.

I suggest the authors, if possible, to update the data.

Reviewer 3 Report

See the attachment.

Round 2

Reviewer 1 Report

Thank the authors for following my comments.

Reviewer 2 Report

thank you for editing the article and making a note regarding the interpretation of the data.

Reviewer 3 Report

Please refer to the attached file
